# Association of Obesity with the Risk of Hyperhomocysteinemia among the Chinese Community Residents: A Prospective Cohort Study in Shanghai, China

**DOI:** 10.3390/nu13103648

**Published:** 2021-10-18

**Authors:** Yu Xiang, Qi Zhao, Na Wang, Yuting Yu, Ruiping Wang, Yue Zhang, Shuheng Cui, Yiling Wu, Xing Liu, Yonggen Jiang, Yongfu Yu, Genming Zhao

**Affiliations:** 1The Key Laboratory of Public Health Safety of Ministry of Education, Department of Epidemiology, School of Public Health, Fudan University, Shanghai 200032, China; xiangy20@fudan.edu.cn (Y.X.); zhaoqi@shmu.edu.cn (Q.Z.); na.wang@fudan.edu.cn (N.W.); 17211020011@fudan.edu.cn (Y.Y.); cuishuheng1995@outlook.com (S.C.); liuxing@fudan.edu.cn (X.L.); 2Songjiang District Center for Disease Control and Prevention, Shanghai 201600, China; ruipingwang@163.com (R.W.); aries2119@163.com (Y.W.); Sjjkzx1106@126.com (Y.J.); 3Department of Epidemiology, School of Public Health, Shanxi Medical University, Taiyuan 030001, China; 17111020009@fudan.edu.cn; 4The Key Laboratory of Public Health Safety of Ministry of Education, Department of Biostatistics, School of Public Health, Fudan University, Shanghai 200032, China; yu@fudan.edu.cn

**Keywords:** obesity, hyperhomocysteinemia, cohort study, Chinese community residents

## Abstract

A prospective community-based cohort study was conducted to investigate the effects of obesity on hyperhomocysteinemia (HHcy) in community residents from Shanghai, China, with a median follow-up period of 2.98 years. The exposures were high body mass index (BMI) (BMI ≥ 28.0 kg/m^2^) and high waist circumference (WC) (WC ≥ 85.0 cm for female and WC ≥ 90.0 for male) at baseline investigation, and the outcome was the incident of HHcy after the follow-up. A restricted cubic spline (RCS) was performed to assess the possible nonlinear relationship of BMI and WC with HHcy. A Cox proportional hazard regression model was used to evaluate the association between BMI and WC measured obesity and the risk of HHcy (Hcy level > 15 µmol/L). No significant non-linearity was found between BMI and WC with HHcy. Cox regression model showed that underweight measured by BMI was negatively associated with the risk of HHcy after controlling for confounder variables (adjusted HR = 0.64, 95% CI = 0.42 to 0.99). While abdominal obesity was positively associated with the risk of HHcy for those without CVD-related comorbidities (adjusted HR = 1.26, 95% CI = 1.05 to 1.51). Our results suggested that individuals could maintain a relatively low BMI and normal WC to lower the risk of HHcy.

## 1. Introduction

Homocysteine (Hcy), an intermediate product in the normal biosynthesis of the amino acids methionine and cysteine, has proven to have adverse effects on cardiovascular endothelium and could cause cardiovascular problems [1]. Cardiovascular disease (CVD), including coronary artery diseases, stroke, and heart attacks, is the leading cause of death in China. The annual number of deaths due to CVD increased from 2.51 million in 1990 to 3.97 million in 2016 in China. In 2016, there were nearly 94 million cases of cardiovascular disease, while the number is still rising in recent years [2]. Hcy has been identified as an important determinant of CVD, and recognized as an independent risk factor for stroke, heart attack and atherosclerosis; moreover, recent studies even demonstrated that Hcy is associated with cancer, Alzheimer’s disease, Parkinson’s disease and autism [3,4]. Hyperhomocysteinemia (HHcy) is defined as a medical condition characterized by high Hcy level in the blood [3]. Evidence has shown that HHcy leads to endothelial cell damage and could reduce the flexibility of vessels. Besides, HHcy could aggravate the adverse effects of risk factors such as hypertension, lipid and lipoprotein metabolism, and the development of inflammation [5]. The prevalence of HHcy may vary between populations significantly and most likely depend on age and genetic background [1].

The risk of CVD could be affected not only by HHcy, but also could be increased by obesity through its influence on the development and severity of comorbidities such as hypertension, dyslipidemia, diabetes, or glucose intolerance [6]. In China, around 46% of adults are obese or overweight, leading obesity to become a major public health crisis among the Chinese population [7]. More importantly, obesity has been recognized as an independent risk factor for the development of CVD, such as heart failure and acute cardiovascular syndromes (ischemic stroke and hemorrhagic stroke) [6,8,9]. In general, obesity is often evaluated based on the body mass index (BMI), which is used as a peripheral measurement instead of analyzing body composition to determine body fat mass. Besides, waist circumference (WC) is also commonly used to assess the condition of abdominal obesity.

To date, the published evidence assessed on the association between obesity and the risk of HHcy in the general population is scarce and restricted [10,11,12], and the evidence with different population settings is conflicting [13,14]. To assess the association between obesity and HHcy to prevent its related potential CVD outcomes, we conducted a community-based prospective cohort study with a median follow-up period of 2.98 years to explore the associations between obesity with different measurements and the risk of HHcy in Chinese adults.

## 2. Materials and Methods

### 2.1. Study Design and Subjects

The study was performed with participants of the ongoing, prospective, population-based cohort study in multiple districts of Shanghai, China. We adopted a multistage, stratified, clustered sampling method to enroll subjects from different communities randomly. Residents aged from 20 to 74 years old and who lived in the four communities (Xinqiao, Sheshan, Maogang, and Zhongshan community) of Songjiang District for at least five years were enrolled in the baseline survey. The initial baseline investigation began in June 2016, which included 35,727 community residents from Songjiang District, Shanghai. A clustered random sampling method was applied to recruit 15,000 subjects to receive the in-person follow-up investigation, which consisted of surveys and anthropometric and biochemical examinations at the local community health centers. For those who did not receive the in-person follow-up, a follow-up based on the local health information system was conducted to capture their major outcome events. Before conducting the in-person follow-up, subjects were contacted by the working staff at the local community health centers to inquire about their willingness to participate. Among 15,000 subjects, there were 12,305 respondents agreed to participate in the in-person follow-up. The response rate to receive the in-person follow-up was 82.0%. The in-person follow-up examinations were conducted from April 2019 to August 2020 in different regions. For all subjects included in our cohort, sociodemographic information, health status by self-evaluation, lifestyle, and the history of chronic diseases were collected using a structured questionnaire by trained interviewers recruited by the community healthcare centers.

After finishing the in-person follow-up with a median follow-up period of 2.98 years, 2779 subjects already had HHcy at baseline investigation and were excluded from the study. We also excluded 339 participants who did not complete anthropometric examinations, 215 participants with missing information on biological blood Hcy index, and 20 participants who did not complete the questionnaire on their sociodemographic information. In total, there were 8952 subjects included in our study. The flowchart of the study participants recruitment was presented in Figure 1. The cohort profile for our study could be accessed elsewhere [15].

### 2.2. Anthropometric and Other Measurements

All subjects in our study received a physical examination that recorded their anthropometric measurements (height, weight, waist circumferences) performed by licensed physicians in the community healthcare centers. Height and weight were measured with subjects wearing light clothing and no footwear, and were accurate to 0.1 cm and 0.1 kg, respectively. Based on the reference standard of the Chinese Body Mass Index, BMI (kg/m^2^) was categorized into four classifications: underweight (<18.5), normal (18.5–23.9), overweight (24.0–27.9), and obesity (>28.0) [16]. Waist circumference (WC) was measured accurately to 0.1 cm at the midpoint between the lower rib and the upper iliac crest, and was categorized into two groups: normal weight (WC < 85.0 cm for female and WC < 90.0 for male) and abdominal obesity (WC ≥ 85.0 cm for female and WC ≥ 90.0 for male) [17]. Subjects were categorized by age as 20–39, 40–49, 50–59, 60–69, or ≥70 years old. The education level was categorized as “Middle school or below” (never received education or completion of 9 years of compulsory education or graduated from the primary school), “High school or above” (completion of a college degree or higher or finished the 12 years of education from high school). Smoking status was determined as “never smoked”, “ex-smoker”, or “current smoke” and then categorized into two categories: “non-current smoker” and “current smoker”. Alcohol drinking status was determined based on the response to the question “Have you ever consumed alcohol at least three times a week for more than six months?”. Exercising was defined as doing physical exercises for at least 10 min every week over the past year. Cardiovascular comorbidities were defined as subjects’ self-reported cardiovascular-related comorbidities such as hypertension, coronary heart disease, stroke, and diabetes. Tea drinking was defined as drinking tea at least three times per week for more than six months. Retired status was defined as whether the subject has retired from work.

### 2.3. Blood Sampling and Measurements

Blood samples (approximately 10 mL for each participant) were collected from an antecubital vein in the morning after a 12 h overnight fast. Samples were stored at −80 °C freezer for no more than 6 h until being transported to the Dian Diagnostics Co. Ltd. (Hangzhou, China) (a medical laboratory center) for further analysis. Serum Hcy was measured using the enzymatic cycling method [18]. Subjects with Hcy levels greater than 15 µmol/L were considered to have HHcy.

### 2.4. Statistical Analysis

The descriptive statistics of categorical variables were presented as frequency and the column percentage. The normality of data was assessed by the Kolmogorov–Smirnov test. Student *t*-test and one-way ANOVA were used to compare continuous data, and $\chi^{2}$ tests were used for categorical data. Dunnett’s multiple comparison test was used to compare the means between different groups. A two-way ANOVA was performed to test for interactions between BMI or WC groups with different genders.

We also used restricted cubic splines (RCS) with five knots at the 5th, 35th, 50th, 65th, and 95th centiles to model the association of BMI and WC within the Cox regression model. In the spline models, baseline BMI and WC were mutually adjusted. Covariates included in RCS models were age, gender, age, education level, alcohol drinking, smoking, tea drinking, exercise, retirement status, CVD-related comorbidities, and baseline blood creatine level to control for their potential confounding effects.

We used Cox proportional hazards regression model to examine the associations between BMI and WC with the risk of HHcy. Schoenfeld individual test was used to test the proportional hazard. BMI and WC were modelled as categorical variables. The regression models included follow-up time as the time scale in months. In the multivariable analysis, we further controlled for age, gender, education level, smoking status, alcohol drinking status, exercise, tea drinking, retired status, CVD-related comorbidities and baseline blood creatine level. The dependent variable was categorized into two groups: HHcy (Hcy level of >15 µmol/L) and Non-HHcy (Hcy level ≤15 µmol/L). The associations between BMI and WC with the risk of HHcy were assessed using hazard ratios (HRs) and 95% confidence intervals (95% CIs). The normal weight groups measured by BMI and WC were considered as the reference group for the Cox regression models. Stratified analyses were conducted according to gender, age, and CVD-related comorbidities status. A two-tailed *p*-level of 0.05 level was considered statistically significant. A post hoc power was calculated for the model 4 of the BMI groups which included all the confounder variables in this study. All statistical analyses were performed using R version 4.0.1.

### 2.5. Ethical Approval

The study was approved by the Institute Review Board (IRB) of the School of Public Health, Fudan University (Authorization number: IRB#2016-04-0586). Written informed consent was obtained from all participants before the launch of the study.

## 3. Results

Table 1 shows the distribution characteristics of the study variables among subjects with or without HHcy after the follow-up examinations. The mean age of the subjects was 57.49 ± 9.79 years old. The median years of follow-up was 2.98 years, and we identified 2104 (23.5%) new incidents of HHcy. The mean BMI was 24.38 kg/m^2^, and the mean WC was 80.99 cm. The mean Hcy of our subjects was 13.57 ± 5.08 µmol/L, with 1268 (44.9%) male and 836 (13.6%) female subjects having Hcy >15 µmol/L respectively. At baseline investigation, there were 3499 (39.16%) subjects underweight, and 1234 (13.81%) subjects were obese based on the measurements of BMI, while 2537 (28.34%) subjects had abdominal obesity measured by WC. Subjects with HHcy had a higher proportion of obesity measured by BMI compared with subjects without HHcy (*p* < 0.001). Factors such as cardiovascular-associated comorbidities, smoking status, alcohol drinking status, tea drinking habits, retired status, educational degree, exercise status are proved to be statistically significant associated with HHcy (*p* < 0.001).

We examined the possible non-linear relationship of BMI and WC with HHcy in all subjects, male and female subjects when controlling for age, gender, education level, alcohol drinking, smoking, tea drinking, exercise, retirement status, CVD-related comorbidities, and baseline blood creatine level. As shown in Figure 2, no significant nonlinear relationship was observed for BMI and WC with HHcy (*p* for non-linearity > 0.05). But we observed a negative association between BMI and the risk of HHcy for all subjects. After dividing BMI and WC into different groups according to the standards, we found that there was a significant difference in the mean serum Hcy level between underweight, overweight, obesity groups and the normal weight group (*p* < 0.001), and also a significant difference in mean Hcy level between WC-measured abdominal obesity group and normal WC group (t = 2.44, *p* = 0.015). However, a two-way ANOVA test showed there were interactions between different BMI or WC groups with gender (For BMI groups and gender: F-value = 7.23, *p* < 0.01; For WC groups and gender: F-value = 12.72, *p* < 0.01). Therefore, we stratified our analysis according to different sex groups. Among the females, the significant results could be noticed in the mean Hcy levels between BMI groups and WC groups. Dunnett’s t multiple comparisons tests showed that the mean serum Hcy levels in the underweight, overweight, and obesity groups are all significantly different from the normal weight group among females (Figure 3). The same significant results could not be observed among the males. For males, the mean Hcy level in the WC-measured obesity group is 16.06 ± 6.22 µmol/L, and there was no significant difference in the mean Hcy level between different BMI or WC groups (BMI groups: F-value = 0.82, *p* = 0.48; WC groups: t = 0.85, *p* = 0.39).

We further examined the association of BMI with the risk of HHcy (Table 2). Compared with those in the normal weight group measured by BMI, subjects in the underweight group had a hazard ratio of 0.57 (95% CI = 0.37 to 0.87) for the risk of HHcy in model 1. After adjusting for all the covariates, the same negative association between the underweight group and the risk of HHcy could also be observed in other different models. The power of our results for the BMI groups in model 4 was calculated to reach 1.0 at a 0.05 significance level to detect a hazard ratio of 0.64, based on the current sample size of 8952, with a multiple regression of the BMI group on the other covariates in our Cox regression model had an R-squared of 0.43. When doing the stratification analysis, this negative association could only be observed in the female subjects (adjusted HR= 0.49, 95% CI: 0.29 to 0.95). However, no association was found between BMI-measured underweight group and obesity group with the risk of HHcy in all subjects or the other stratifications, except for those with CVD-related comorbidities, overweight measured by BMI was negatively associated with the risk of HHcy (adjusted HR= 0.84, 95% CI: 0.71 to 0.98). In Table 3, when assessing the association between WC-measured obesity and the risk of HHcy, we found that only adjusted for age and gender, abdominal obesity measured by WC was positively associated with HHcy (adjusted HR = 1.11 (95% CI: 1.01 to 1.21)). After adjusting for other covariates included in the model 3 and 4, this positive association faded away. For subjects without CVD-related comorbidities, a positive association for HHcy from abdominal obesity could also be observed (adjusted HR = 1.26 (95% CI: 1.05 to 1.51)).

## 4. Discussion

In this large prospective cohort study, we used validated anthropometric data to examine the association of BMI and WC with the risk of HHcy. We found a negative association between the underweight group measured by BMI and the risk of having HHcy after controlling for confounder factors. In contrast, abdominal obesity measured by WC was positively associated with HHcy in subjects without CVD-related comorbidities.

Our findings on the relationship between BMI and serum Hcy concentration are consistent with those reported by others [14,19]. In contrast, studies stated that there was a positive association between BMI or WC and HHcy [12,20,21]. However, neither the study design nor the characteristics of subjects of studies with opposite conclusions were similar to ours. Besides, several studies indicated no associations between BMI or waist circumference and plasma Hcy concentration but found that BMI is related to methylation reactions [22,23,24,25,26]. Therefore, further studies investigating the association between BMI and Hcy metabolism are warranted.

Although we found a negative association between the underweight group and the normal weight group with their serum Hcy concentration, its mechanisms remained unknown. Though Hcy is found in high concentrations in the serum of overweight and obese females, which could trigger the immune response by hypermethylation of some specific gene promoters and chelate divalent metal ions such as ${\mathrm{Cu}}^{2+}$ and ${\mathrm{Zn}}^{2+}$, but this role of Hcy has not been established in relationship with obesity [27]. The deficiency of vitamins B6, B12, and folic acid could lead to the development of hyperhomocysteinemia. Vitamin B6 plays an essential role in the catabolism of Hcy into cysteine, and vitamin B12 could elicit the methylation process of Hcy into methionine [28]. Patients diagnosed with abdominal obesity, despite daily high-energy consumption in general, are generally observed to have vitamin and mineral deficiencies, which is associated with the low quality of the food they consume. These nutritional deficiencies are common in the obese population, resulting in the abnormal level of plasma folate and Hcy concentration [29]. Furthermore, future studies are needed to elucidate the mechanisms from the viewpoint of the observed positive association between abdominal obesity and the risk of HHcy.

The known risk factors for HHcy are nutritional deficiency (lack of folate, vitamin B6, or vitamin B12), genetic variations (default in cystatin B synthetase or 5,10-methylenetetrahydrofolate reductase), impaired kidney function, certain carcinomas, hypothyroidism, and smoking [30]. Our study suggests that abdominal obesity measured by WC could be another independent determinant of HHcy for those without CVD-related comorbidities.

It has been proved that abdominal obesity was a vital risk factor of insulin-resistance (IR) syndrome [31]. IR, typically defined as decreased sensitivity and responsiveness to insulin-mediated glucose disposal, and inhibition of hepatic glucose production, could impair muscle cells’ ability to take up and store glucose and triglycerides. As a result, the glucose and triglycerides circulating in the blood are in high levels [32,33]. IR is commonly prevalent in older adults but recently has become more common at all ages, including middle-aged adults who are overweight and live sedentary life [33]. Moreover, IR is associated with a metabolic and cardiovascular cluster of disorders (hypertension, visceral obesity, endothelial dysfunction, etc.), which are independent risk factors for CVD [34]. It has been proven by other researchers that there was a significant association between insulin level and Hcy concentration, which could link metabolic syndrome with Hcy [35]. Although we did not observe any positive association between obesity measured by BMI and the risk of HHcy, the underlying mechanism for why female subjects with obesity or overweight measured by BMI had significant higher plasma Hcy level compared with the normal weight group, could be due to this linkage between IR level and metabolic disorders.

In recent years, researchers have found that obesity has a potentially protective effect when it coexists with CVD, including heart failure, coronary heart disease, hypertension, atrial fibrillation, pulmonary arterial hypertension [36]. This phenomenon may represent a “lean paradox”, resulting in the progressive catabolic state and lean mass loss, leading normal weight or underweight individuals to have a poorer prognosis in CVD cases compared with patients classified as severe or morbidly obese. Increasing cardiorespiratory fitness (CRF) could lower CVD risk, in spite of obesity classified by BMI. Therefore, body composition assessment (i.e., fat mass, fat-free mass, lean body mass) could be a better indicator of CVD than BMI alone [37]. It could be reflective in our study that more accurate measurements to assess human body composition should be taken to further assess whether individuals with higher lean body mass, but higher BMI could have a lower risk of HHcy.

Whenever the Hcy concentration elevates in the body, whether genetic or acquired, it could lead to the generation of multiple diseases. Hcy is positively associated with CVDs, and could be an in dependent risk factor for neurodegenerative diseases, such as Alzheimer’s disease, Parkinson’s disease, and autism. Although obesity has been identified as an independent factor for CVD, the linkage between obesity and HHcy has not been well established. The public health implication for this study is that the early intervention of abdominal obesity on the community level to prevent HHcy and its related consequences is important for community healthcare practitioners [38]. To prevent HHcy and its further consequences, public health practitioners could recommend that individuals could maintain a relatively low BMI and normal WC instead of having abdominal obesity.

The strength of this article is its community-based prospective cohort study setting. The population sample is large, and the subjects that participated in this study were active. The effect of reverse causation could be mitigated within this study design. In addition, whereas some studies examined Hcy levels in Chinese individuals with regard to lifestyle factors and chronic diseases [14,39,40,41], this is the first to show descriptive statistics to assess BMI and WC and the risk of HHcy in Chinese adult residents of the community.

The weaknesses of our study are that we did not collect detailed information on subjects’ vitamin, folate intake status, and micro-nutrient substances such as ${\mathrm{Cu}}^{2+}$ and ${\mathrm{Zn}}^{2+}$; and these could also be the confounders in our study. Besides, baseline characteristics such as comorbidities officially diagnosed by the medical organizations instead of self-reported by subjects (e.g., Alzheimer’s, Parkinson’s Disease, etc.) are not captured. In addition, although the post-hoc power for our analysis in model 4 was adequate to detect the hazard ratio for the underweight group, we did not perform prior power analysis to calculate the probability of detecting a certain effect size within the certain sample size and significance level, which should be improved in our future analysis.

As noted above, these issues are essential to be clarified in future research. Besides, as the subjects in our study were from the Songjiang District of Shanghai, a suburban area in East China, and this could limit the generalizability of our study results; thus, we might not estimate the incidence of HHcy among adults in Shanghai. Moreover, we should take more accurate measurements to assess the human body fat mass, fat-free mass, and lean body fat in the future.

## 5. Conclusions

In summary, we found a negative association of underweight measured by BMI with the risk of having HHcy after controlling for confounder factors. In contrast, abdominal obesity measured by WC was positively associated with the risk of HHcy in subjects without CVD-related comorbidities. Our results suggested that individuals could maintain a relatively low BMI and normal WC instead of having abdominal obesity to lower the risk of HHcy.

## Figures and Tables

**Figure 1 nutrients-13-03648-f001:**
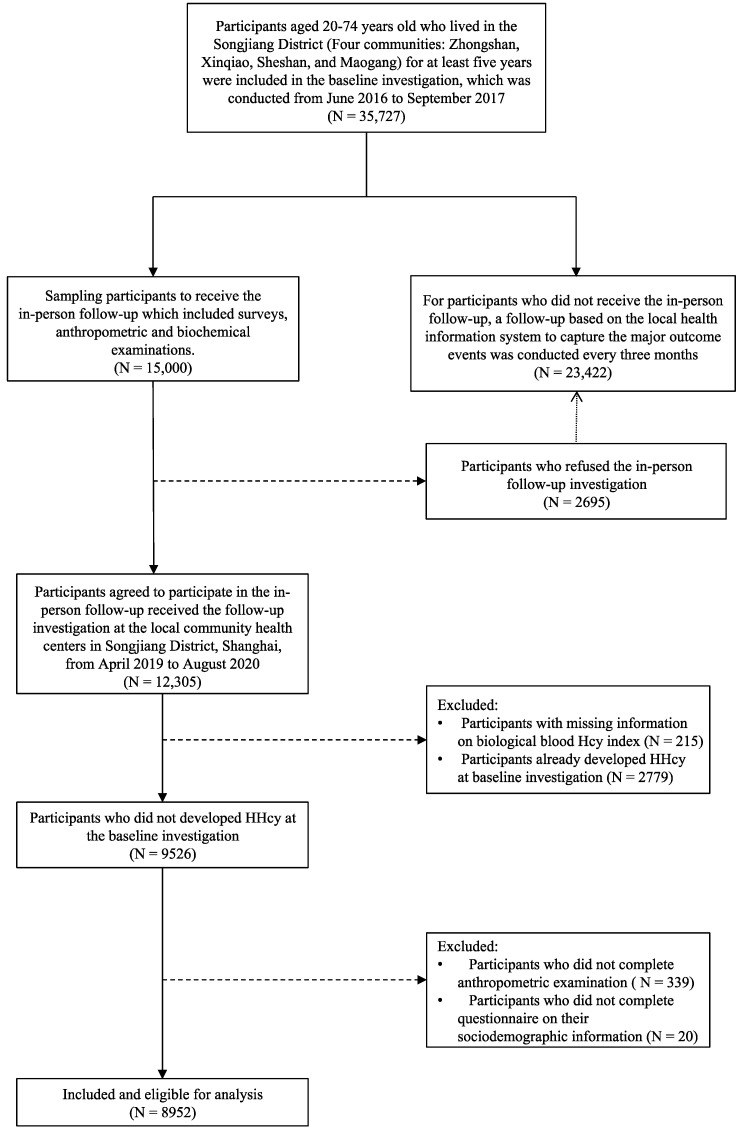
Flowchart of the study participants recruitment.

**Figure 2 nutrients-13-03648-f002:**
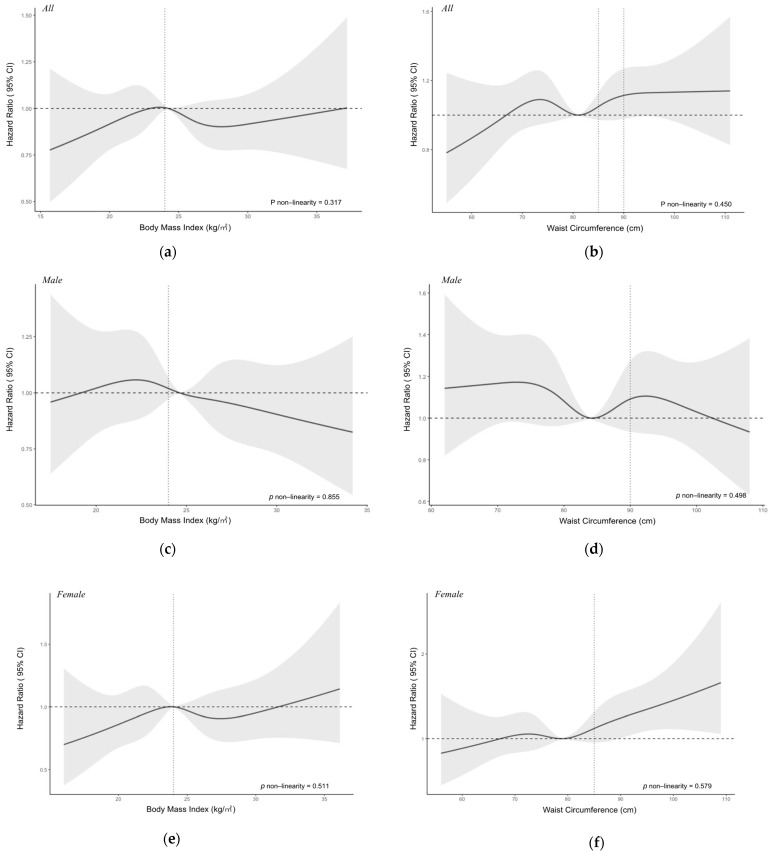
Nonlinear relationship of body mass index (BMI) and waist circumference (WC) with hyperhomocysteinemia among all subjects (**a**,**b**), male (**c**,**d**) and female subjects (**e**,**f**). Adjusted for age, gender, education, alcohol drinking, smoking, tea drinking, exercise, retirement status, cardiovascular disease comorbidities, and baseline blood creatine level. Y-axis represents the adjusted hazard ratio for the risk of Hyperhomocysteinemia for any value of BMI or WC. The black solid line represents the hazard ratio, and the grey shaded area represents the 95% confidence intervals. The horizontal dotted line represents the reference hazard ratio line y = 1. The vertical dotted line for the left graphs represents the cut-off value for BMI and WC.

**Figure 3 nutrients-13-03648-f003:**
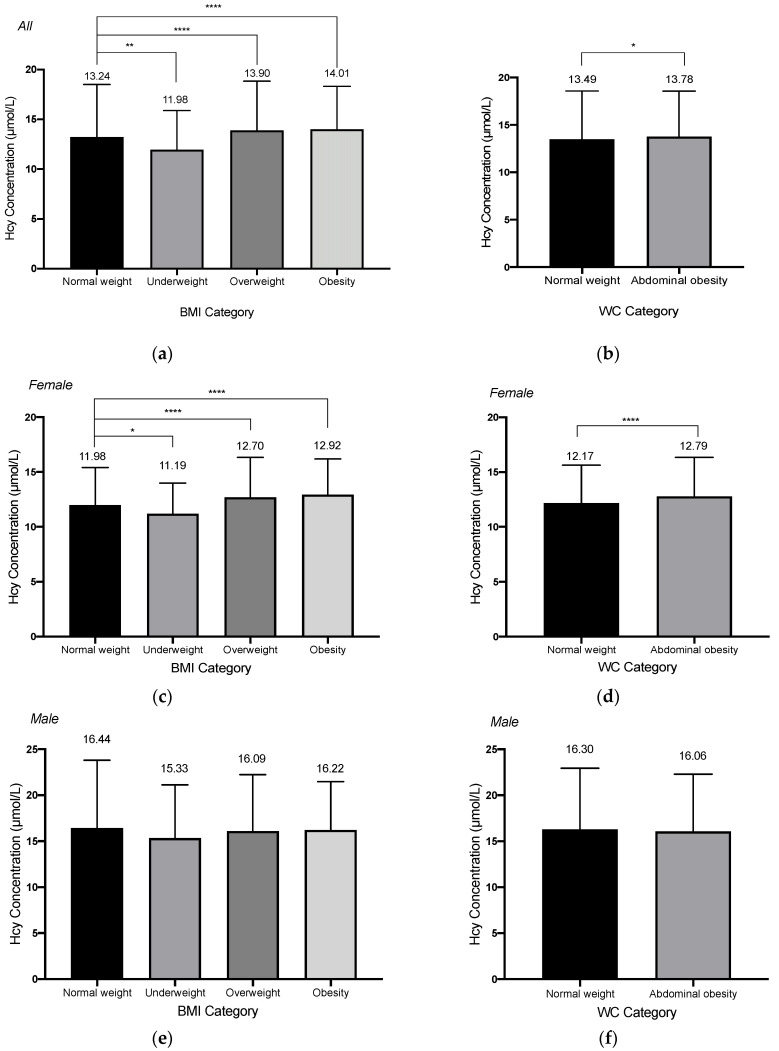
The homocysteine concentrations categorizing by body mass index (BMI) or waist circumference (WC) for all subjects (**a**,**b**), female subject (**c**,**d**) and male subjects (**e**,**f**). For BMI categories, subjects were categorized into four categories (underweight: <18.5 kg/m^2^; normal weight: 18.5–23.9 kg/m^2^; overweight: 24.0–27.9 kg/m^2^; obesity: ≥28.0 kg/m^2^); for WC categories, subjects were categorized into two categories (Normal weight: <85.0 cm for female, <90.0 cm for male; abdominal obesity: ≥85.0 cm for female, ≥90.0 for male). * 0.01 < *p* < 0.05; ** 0.001 < *p* < 0.01; **** *p* < 0.0001.

**Table 1 nutrients-13-03648-t001:** Distributions of study variables among subjects with or without hyperhomocysteinemia (HHcy).

	Hhcy	Non-Hhcy	Total	*p* Value
	*n* = 2104	*n* = 6848	*n* = 8952	
Age (years)				<0.001
20–39	41 (1.95)	521 (7.61)	562 (6.28)	
40–49	95 (4.52)	867 (12.66)	962 (10.75)	
50–59	524 (24.90)	2556 (37.32)	3080 (34.41)	
60–69	1145 (54.42)	2497 (36.46)	3642 (40.68)	
≥70	299 (14.21)	407 (5.94)	706 (7.89)	
Body mass index (kg/m^2^)				<0.001
Normal Weight	829 (39.51)	3168 (46.34)	3997 (44.74)	
Underweight	22 (1.05)	182 (2.66)	204 (2.28)	
Overweight	902 (42.99)	2597 (37.99)	3499 (39.16)	
Obese	345 (16.44)	889 (13.00)	1234 (13.81)	
Waist circumference (cm)				0.017
Normal	1464 (69.58)	4951 (72.30)	6415 (71.66)	
Abdominal obesity	640 (30.42)	1897 (27.70)	2537 (28.34)	
Gender				<0.001
Male	1268 (60.27)	1552 (22.66)	2820 (31.50)	
Female	836 (39.73)	5296 (77.34)	6132 (68.50)	
Education Degree				<0.001
Middle School or Below	1264 (60.54)	3456 (51.30)	4720 (53.48)	
High School or Above	824 (39.46)	3281 (48.70)	4105 (46.52)	
Cardiovascular comorbidities				<0.001
Ye	1024 (48.67)	2489 (36.35)	3513 (39.24)	
No	1080 (51.33)	4359 (63.65)	5439 (60.76)	
Alcohol Drinking				<0.001
Yes	448 (21.29)	629 (9.19)	1077 (12.03)	
No	1656 (78.71)	6219 (90.81)	7875 (87.97)	
Smoking				<0.001
Yes	727 (34.55)	903 (13.19)	1630 (18.21)	
No	1377 (65.45)	5945 (86.81)	7322 (81.79)	
Tea Drinking				<0.001
Yes	846 (40.21)	1531 (22.36)	2377 (26.55)	
No	1258 (59.79)	5317 (77.64)	6575 (73.45)	
Retirement				<0.001
Yes	1613 (76.66)	4308 (62.91)	5921 (66.14)	
No	491 (23.34)	2540 (37.09)	3031 (33.86)	
Exercise				<0.001
Yes	816 (38.78)	2302 (33.62)	3118 (34.83)	
No	1288 (61.22)	4546 (66.38)	5834 (65.17)	

**Table 2 nutrients-13-03648-t002:** Hazard ratios (HRs) and 95% confidence intervals (CIs) of HHcy according to Body Mass Index (BMI).

	Normal Weight 18.5 ≤ BMI ≤ 23.9		Underweight BMI < 18.5		Overweight 24.0 ≤ BMI ≤ 27.9		Obesity BMI ≥ 28		Total
	Cases/*n*	HR (ref)	Cases/*n*	HR (95% CI)	Cases/*n*	HR (95% CI)	Cases/*n*	HR (95% CI)	
**All** **subjects *^a^***	829/3997		22/204		902/3499		829/3997		8934
Model 1		1.0		0.57 (0.37 to 0.87) **		1.07 (0.98 to 1.18)		1.10 (0.97 to 1.24)	
Model 2		1.0		0.64 (0.42 to 0.98) *		1.02 (0.93 to 1.12)		1.06 (0.93 to 1.20)	
Model 3		1.0		0.63 (0.40 to 0.96) *		0.99 (0.89 to 1.10)		1.04 (1.91 to 1.19)	
Model 4		1.0		0.64 (0.42 to 0.99) *		0.95 (0.85 to 1.07)		0.97 (0.82 to 1.15)	
**Gender *^b^***									
Male	503/1131	1.0	13/49	0.82 (0.46 to 1.45)	560/1237	0.92 (0.80 to 1.07)	190/407	0.88 (0.70 to 1.11)	2814
Female	326/2866	1.0	9/165	0.49 (0.29 to 0.95) *	342/2262	1.01 (0.84 to 1.20)	155/827	1.08 (0.82 to 1.40)	6120
**Age** **(years)** ** * ^c^ * **									
20–39	21/331	1.0	1/44	0.66 (0.33 to 4.93)	13/126	0.69 (0.33 to 1.43)	6/60	1.40 (0.55 to 3.50)	561
40–49	37/533	1.0	1/21	2.47 (0.32 to 19.05)	37/302	1.10 (0.61 to 1.97)	20/106	0.94 (0.37 to 2.39)	962
50–59	184/1355	1.0	5/69	0.78 (0.32 to 1.93)	229/1205	0.95 (0.75 to 1.19)	103/440	1.01 (0.72 to 1.41)	3069
60–69	464/1476	1.0	9/52	0.69 (0.35 to 1.35)	500/1583	0.89 (0.77 to 1.04)	169/525	0.81 (0.64 to 1.02)	3636
≥70	123/302	1.0	12/18	0.52 (0.21 to 1.28)	123/283	1.03 (0.75 to 1.41)	47/103	1.13 (0.71 to 1.82)	706
**CVD comorbidities** ** * ^b^ * **									
Yes	325/1159	1.0	3/29	0.32 (0.10 to 1.02)	456/1566	0.84 (0.71 to 0.98) *	238/754	0.86 (0.69 to 1.08)	3508
No	504/2838	1.0	19/175	0.80 (0.50 to 1.29)	446/1933	1.04 (0.90 to 1.23)	107/480	1.10 (0.83 to 1.46)	5426

*^a^* Model 1 adjusted for age only. Model 2 adjusted for age and gender. Model 3 adjusted for age, gender, education, cardiovascular comorbidities, smoking, alcohol drinking, exercising, tea-drinking, retire status and baseline blood creatine level. Model 4 adjusted for all the variables in model and waist circumference additionally. *^b^* Covariables included in the selected stratification were education level, exercising, smoking, tea drinking, alcohol drinking, CVD related comorbidities, retire status, baseline blood creatine level, and baseline waist circumference. *^c^* Covariable adjusted in the age 20–39 group was only gender due to lack of sample size of the stratification group. Covariables included in other selected age stratification groups were the same as Model 4. * 0.01 < *p* < 0.05; ** *p* < 0.01.

**Table 3 nutrients-13-03648-t003:** Hazard ratios (HRs) and 95% confidence intervals (CIs) of HHcy according to Waist Circumference (WC).

	Normal Weight		Abdominal Obesity		Total
	WC < 85.00 cm for Females		WC ≥ 85.00 cm for Females		
	WC < 90.00 cm for Males		WC ≥ 90.00 cm for Males		
	Cases/*n*	HR (95%CI)	Cases/*n*	HR (95%CI)	
**All Subjects *^a^***	1464/6415		640/2537		8952
Model 1		1.0		1.05 (0.96 to 1.15)	
Model 2		1.0		1.11 (1.01 to 1.21) *	
Model 3		1.0		1.10 (0.99 to 1.21)	
Model 4		1.0		1.11 (0.99 to 1.24)	
**Gender *^b^***					
Male	925/2051	1.0	343/769	1.09 (0.93 to 1.27)	2820
Female	539/4364	1.0	297/1768	1.09 (0.91 to 1.30)	6132
**Age (years) *^c^***					
20–39	35/489	1.0	6/73	0.93 (0.38 to 2.25)	562
40–49	69/783	1.0	26/179	1.20 (0.60 to 2.42)	962
50–59	372/2274	1.0	152/806	1.19 (0.59 to 2.43)	3080
60–69	793/2422	1.0	352/1220	1.15 (0.98 to 1.34)	3642
≥70	195/447	1.0	104/259	0.93 (0.67 to 1.28)	706
**CVD comorbidities** ** * ^b^ * **					
Yes	623/2098	1.0	401/1415	1.08 (0.93 to 1.27)	3513
No	841/4317	1.0	239/1122	1.26 (1.05 to 1.51) *	5439

*^a^* Model 1 adjusted for age only. Model 2 adjusted for age and gender. Model 3 adjusted for age, gender, education, cardiovascular comorbidities, smoking, alcohol drinking, exercising, tea-drinking, retire status and baseline blood creatine level. Model 4 adjusted for all the variables in model and body mass index additionally. *^b^* Covariables included in the selected stratification were education level, exercising, smoking, tea drinking, alcohol drinking, CVD related comorbidities, retire status, baseline blood creatine level, and baseline body mass index. *^c^* Covariable adjusted in the age 20–39 group was only gender due to lack of sample size of the stratification group. Covariables included in other selected age stratification groups were the same as Model 4. * 0.01 < *p* < 0.05.

## Data Availability

The dataset used and analyzed during the current study is available from the corresponding author upon reasonable request.

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
