# Peer review of "Association of Obesity with the Risk of Hyperhomocysteinemia among the Chinese Community Residents: A Prospective Cohort Study in Shanghai, China"

_nutrients, 2021, doi:10.3390/nu13103648_

Round 1

Reviewer 1 Report

The current manuscript by Xiang et al. provides results from a prospective community-based cohort study to investigate the effects of obesity on Hyperhomocysteinemia (HHcy) in community residents from Shanghai, China. The authors have elegantly described their findings and very well outlined the strength and weaknesses of the study. The main advantage of the study is the population sample size which provides a level of confidence in conclusions made by authors.

Author Response

Dear reviewer,

Thank you for your helpful and pertinent comments and suggestions for our paper. We have revised the manuscript according to your suggestions. 

In line with your instructions, we provide below a numbered point-by-point response to each comment.

 Point 1: The current manuscript by Xiang et al. provides results from a prospective community-based cohort study to investigate the effects of obesity on Hyperhomocysteinemia (HHcy) in community residents from Shanghai, China. The authors have elegantly described their findings and very well outlined the strength and weaknesses of the study. The main advantage of the study is the population sample size which provides a level of confidence in conclusions made by authors.

Response 1: We appreciate the Reviewer’s positive comments. We now revised our English language and style, and checked for our English spell according to your comments.

We are truly appreciated for your comments!

We believe that the manuscript has improved considerably and hope you will find it suitable for publication in Nutrients. However, we are willing to make further changes if needed.

Sincerely,

Genming Zhao, MD, PhD

Professor of Epidemiology

Fudan University School of Public Health

131 Dong An Road, Shanghai 200032, PRC

Tel: 86-21-54237334

E-mail: gmzhao@shmu.edu.cn

Reviewer 2 Report

Below are my comments

Introduction

  1. Your transition from talking about HHcy to obesity was abrupt. A better transition sentence and a new paragraph where you talk about how obesity is also a risk factor of CVD and then provide evidence would be a much better approach.

Methodology

  1. Out of how many potential participants did you recruit 12,305?
  2. Did you perform an a priori power analysis?
  3. How were subjects recruited? How were they contacted? What percentage of respondents agreed to participate in the study?
  4. What was the inclusion/exclusion criteria of the study besides being between the ages of 20-74 and living in Shanghai for a minimum of 5 years? Why did you omit younger and older people?
  5. I think your citations might be off because study 13 was published in 2006, while your first wave of data collection didn't start until 2016.
  6. Was all the data collected in 1 sitting? Multiple sessions?
  7. I understand that you ran ANOVAs and t-tests on continuous variables (i.e. Hcy, BMI). I would recommend controlling for sex in your group analyses. Based on what you report in Table 1, your analyses might falsely show higher BMI because there are more men in the high Hcy group. Although I can't tell from your tables, that might also be true for the different weight categories.

Results

  1. Overall the results are well written however, I question whether the ANOVA/t-test results are truly giving you what you're looking for or whether the large sex differences is what's accounting for some of these significant results.
  2. Can you please report post-hoc power of your models. Your underweight model with so few n and so many confounders that you are controlling for are the models I am most interested in understanding whether it is adequately powered or not. I think the others should be fine, but it would help if you reported post-hoc power.

Discussion

While you did a good job with this study when you talked about vitamin deficiency I was left wondering why you were talking about that. One of the things that you should mention in your discussion is the fact that you did not collect micronutrient data however, there is a relationship between micronutrients and obesity and Hcy. 

Round 2

Reviewer 2 Report

I believe that the authors did a great job of addressing most of my comments.

The only minor comment I have is to please report calculated power in the results for Model 4 for all of your BMI groups. This may help the reader better interpret the findings. 

It is okay to have a low powered group, especially in an observational study since you cannot control who you recruited. I believe just providing that information and addressing it like you have in your limitations is fine. However, providing calculated power make the findings easier to interpret. 
